# Staging Liver Fibrosis with Hepatic Perivascular Adipose Tissue as a CT Biomarker

**Skylar Chan**[*][1]                                     SPENCER.CHAN@NIH.GOV
**Tejas Sudharshan Mathai**[*][1]                          TEJAS.MATHAI@NIH.GOV
**Praveen T.S. Balamuralikrishna**[1]              THOPPEYSRINIVP2@NIH.GOV
**Vivek Batheja**[1]                                    VIVEK.BATHEJA@NIH.GOV
**Jianfei Liu**[1]                                          JIANFEI.LIU@NIH.GOV
**Meghan G. Lubner**[2]                              MLUBNER@UWHEALTH.ORG
**Perry J. Pickhardt**[2]                          PPICKHARDT2@UWHEALTH.ORG
**Ronald M. Summers**[2]                                        RMS@NIH.GOV

[1] *Radiology and Imaging Sciences, National Institutes of Health Clinical Center, USA*
[2] *Department of Radiology, University of Wisconsin School of Medicine & Public Health, USA*

**Editors:** Accepted for publication at MIDL 2025

## Abstract

Cirrhosis is the $12^{th}$ leading cause of death in the US. There are several CT imaging signs of late fibrosis, such as redistribution of liver segment volume, increased liver nodularity, and periportal space widening. Timely intervention can reverse the progression of early hepatic fibrosis, but later stages are irreversible. We hypothesize that the perivascular adipose tissue (PVAT) around the portal vein arising from periportal space widening may also be predictive of liver fibrosis. In this work, a fully automated pipeline was developed to segment the liver, spleen, portal vein and its branches. The PVAT in the vicinity of the portal vein was identified. From these structures, CT imaging biomarkers (volume, attenuation, fat fraction) were computed. They were used to build uni- and multivariate logistic regression models for diagnosing advanced fibrosis and cirrhosis. The best multivariate model for cirrhosis achieved 93.3% AUC, 78.9% sensitivity, and 93.4% specificity. For advanced fibrosis, the multivariate model obtained 88.7% AUC, 84.2% sensitivity, and 73.7% specificity. The automated approach may be useful for population-based studies of metabolic disease and opportunistic screening.

**Keywords:** CT, Liver Fibrosis, Cirrhosis, Perivascular Adipose Tissue, Portal Vein, Hepatic Arteries

## 1. Introduction

In the US, ∼4.5 million adults are affected by chronic liver disease (Centers for Disease Control and Prevention, 2024). It can lead to chronic inflammation, liver fibrosis, and eventually cirrhosis (Ludwig et al., 2021), which is the $12^{th}$ leading cause of death in the US. Chronic hepatitis B/C viral infection, alcohol abuse, and metabolic dysfunction-associated steatohepatitis (e.g., due to obesity or diabetes) can cause fibrosis. Timely intervention can reverse early fibrosis, and thus it is necessary to distinguish later stages (advanced fibrosis and cirrhosis) from earlier stages of fibrosis.

---

[*] Contributed equally

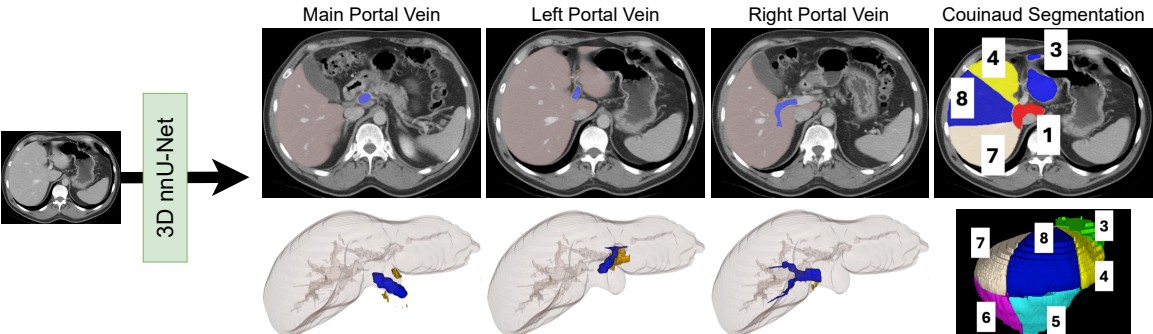

Figure 1: Automated framework for staging advanced fibrosis and cirrhosis. A 3D full resolution nnU-Net segmented the portal vein and its branches. CT biomarkers (volume, attenuation, fat fraction) of the liver, eight hepatic Couinaud segments, spleen and perivascular adipose tissue (PVAT) were automatically derived and used for staging fibrosis. Liver (light brown), portal vein (blue), PVAT (brown), and Couinaud segments (various colors).

Liver biopsy is the gold standard for staging fibrosis, but it is invasive and suffers from sampling error. Lab serum tests can be used, but they are insufficient and cannot replace histology (Pickhardt et al., 2016). Thus, non-invasive imaging-based biomarkers that perform better than biopsy or lab tests are sought (American Liver Foundation, 2022). Certain findings on abdominal contrast-enhanced CT (CECT) clearly indicate liver fibrosis, such as splenomegaly or ascites (Yin et al., 2021; Pickhardt et al., 2017), segmental redistribution (Lee et al., 2022), increased liver surface nodularity (Pickhardt et al., 2016; Mazumder et al., 2023; Mathai et al., 2024a,b; Lewis et al., 2024), and periportal space widening with cavernous transformation (Ludwig et al., 2021; Karcaaltincaba et al., 2007).

Periportal space widening is a manifestation of central liver atrophy (Ito et al., 2000). Due to an increase in distance between the main portal vein and the central liver (Ludwig et al., 2021), fat can expand into the periportal space. As the portal vein supplies 70-80% of hepatic blood flow (Ziessman et al., 2006), there may be local paracrine effects of fat accumulation on nearby regions (Kahn and Bergman, 2022). Recent works suggest that perivascular adipose tissue (PVAT), or the fat surrounding the blood vessels, can contribute to metabolic diseases via a distinct pathway (Kahn and Bergman, 2022; Valentini et al., 2023; Lastra and Manrique, 2015). However, there are very few studies investigating the relationship between fibrosis and PVAT accumulation around vessels feeding and draining the liver (Song et al., 2024; Ludwig et al., 2021; Ito et al., 2000).

In this paper, we hypothesize that PVAT biomarkers surrounding the portal vein in the liver can be used to stage fibrosis. To that end, a fully automated deep learning-based pipeline was developed to segment the liver, spleen, portal vein and its branches. From these segmentations, PVAT and other CT imaging biomarkers were automatically computed. Uni- and multivariate models were built to differentiate advanced fibrosis and cirrhosis from earlier stages. To our knowledge, we are the first to introduce fine-grained segmentations of the individual branches of the portal vein, and to quantitatively examine the utility of perivascular fat around the portal vein for staging liver fibrosis.

## 2. Methods

### 2.1. Patient Sample

In this retrospective study, three datasets containing abdominal contrast-enhanced CT (CECT) scans were used: (1) the public Medical Segmentation Decathlon (MSD) Hepatic Vessels dataset (Task-8) (Antonelli et al., 2022), (2) an internal National Institutes of Health (NIH) dataset, and (3) an external dataset from the University of Wisconsin (UW).

In the MSD dataset, 157 portal venous CT volumes (157 patients) were selected from the 303 volumes in the training subset, based on the clear visualization of the portal vein for annotation. Volume dimensions were $512 \times 512$ voxels in the axial plane, with the number of slices and spacing ranging between (26 - 177) and (1.5 - 8) mm, respectively.

The internal NIH dataset contained 43 patients with cirrhosis and other metabolic diseases having ascites (25 patients) and splenomegaly (18 patients) imaged at the NIH. Volume dimensions were $512 \times 512$ voxels in the axial plane, with the number of slices and spacing ranging between (51 - 680) and (1 - 5) mm, respectively. This dataset was combined with the MSD dataset (157 CT volumes) to yield a total of 200 CT volumes for model training.

In the external UW dataset, 480 patients (304 men, median age: $49 \pm 9$ years) underwent abdominal contrast-enhanced CT exams between 2000 – 2016 (Lee et al., 2022). Each patient underwent one exam and 480 portal venous CT volumes were chosen. The METAVIR staging system was used to categorize patients into 3 groups: patients who underwent CT imaging as potential kidney donors without any known symptoms of liver disease (F0, n = 151); patients with variable degrees of pre-cirrhotic hepatic fibrosis including early (F1, n = 52), intermediate (F2, n = 82), and advanced (F3, n = 56) fibrosis; and patients with chronic liver disease (cirrhosis) who had undergone evaluation for liver transplant (F4, n = 139). Within 1 year of the CT exam, a liver biopsy was required for patients in the early (F1), intermediate (F2), and advanced (F3) fibrosis cohorts.

Patients had the following causes of liver disease: chronic hepatitis B/C viral infection, alcoholism, biliary cirrhosis, sclerosing cholangitis and metabolic dysfunction-associated steatotic liver disease among others. Lab test results (FIB-4 and APRI scores) were also available. A variety of CT scanners were used (GE, Canon, and Siemens). The tube voltage ranged between 100 - 140 kVp with patient specific tube current settings. Volume dimensions were $512 \times 512$ voxels in the axial plane, with the number of slices ranging from 73 to 482 and the spacing varying between 2.5 to 5 mm.

### 2.2. Reference Standard

For the 157 MSD CT volumes, liver and spleen segmentations were obtained using the publicly available TotalSegmentator (TS) tool (Wasserthal et al., 2023). All vessels, including portal veins, hepatic arteries, and hepatic veins, were originally annotated and available as a single label in this dataset. To address this, the portal vein and its branches (left and right) were manually annotated by a research fellow. Two physicians (2+ years of experience) verified and corrected the segmentations of all structures (liver, spleen, portal vein and its branches). The verified portal vein annotation was then separated into its own label and removed (masked out) from the original label, which was subsequently limited to

only hepatic arteries and veins. The rationale for this step stemmed from the challenges in distinguishing the hepatic arteries from hepatic veins on CT.

For the 43 NIH CT volumes, TS was again used to segment the liver and spleen. The hepatic vessels were segmented using the public nnU-Net model (Task-8) (Isensee et al., 2021). The same research fellow annotated the portal vein and its branches, and the same physicians verified the annotations of the liver, spleen, portal vein and its branches.

For the external UW dataset, five patients from each fibrosis stage were randomly chosen to evaluate the segmentation performance of the portal vein. The same physicians manually verified the research fellow's annotation of the portal vein in the 25 UW CT volumes.

### 2.3. Deep Learning Model

Fig. 1 shows the overall framework. A 3D full-resolution nnU-Net was trained with the MSD and NIH datasets and tested on the external UW dataset. The following labels were used for training: liver, spleen, portal vein and its branches, and remaining vessels (hepatic arteries and veins). The primary goal was to segment the portal vein and its branches in the UW dataset. The segmentations of hepatic arteries and veins were discarded at inference.

The nnU-Net model is the de facto standard for segmentation due to its award-winning performance on many tasks (Isensee et al., 2021), such as multi-organ segmentation in CT and MRI. It has often outperformed other architectures, such as transformers-based approaches (Isensee et al., 2024). Thus, only the 3D nnU-Net model was used with no other comparisons. The rationale for nnU-Net originated from the clinical validation goal of this work, which is to extract novel PVAT biomarkers and use it for diagnosing fibrosis.

The nnU-Net framework automatically determined a dataset "fingerprint", which included intensity normalization and resampling to a consistent spacing. Optimal hyper-parameters were automatically computed and no changes were made to the default values. The following training parameters were set: 6 stages with 2 convolution layers per stage, feature maps per stage were [32, 64, 128, 256, 320, 320], kernel sizes were [1, 3, 3, 3, 3, 3], batch size of 2, learning rate of 0.01, SGD optimizer, and 1000 training epochs. The model learned to segment the target structures and iteratively refined the predictions during training via a loss function, which was a combination of the Dice loss and the binary cross entropy loss. Training was conducted on one NVIDIA A100-SXM4-40GB GPU.

### 2.4. Automated Extraction of CT Biomarkers

**Liver and Spleen Biomarkers:** The liver and spleen segmentations in the UW dataset were obtained using TS. A previously validated tool (Lee et al., 2022) delineated the eight hepatic Couinaud segments. This tool was a standard U-Net model trained on a combination of public and private datasets that contained clinician-verified annotations of the 8 hepatic segments. It was validated on two datasets containing patients with various fibrosis levels, and achieved a Dice score exceeding 0.91 for the identification of Couinaud segments (Lee et al., 2022). Using these segmentations, CT biomarkers were computed and they included: volume, attenuation (mean and std. dev.), liver segmental volume ratio (LSVR, the volume ratio of Couinaud segments 1-3 to 4-8) (Lee et al., 2022), and liver surface nodularity (LSN) (Mathai et al., 2024a,b; Lewis et al., 2024).

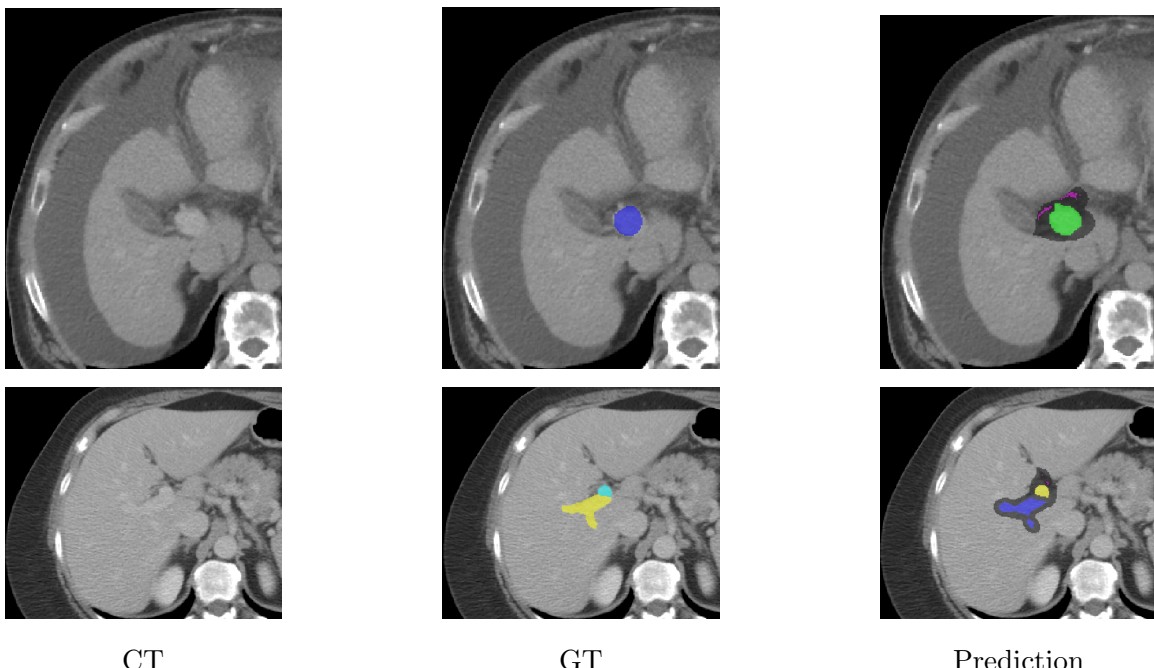

CT           GT           Prediction

Figure 2: Top row is for a cirrhotic patient with ascites, and bottom row is for a normal patient. Last column shows the predicted segmentations of the main portal vein (green), right branch (blue) and left branch (yellow). The perivascular region (dark gray) was extracted and PVAT voxels (magenta) in the [-190, -30] range were identified.

**PVAT Biomarkers:** The portal vein segmentation was dilated with a kernel size of $4\times4\times4$ voxels following prior research (Nguyen et al., 2024; Chatterjee et al., 2022). Other kernel sizes ($3\times3\times3$, $5\times5\times5$) were also tested and no qualitative difference was found between them. The variability in voxel spacing for the UW dataset motivated the choice to use the same kernel size in all dimensions. Resampling all CT volumes to consistent dimensions would introduce artifacts (volume averaging and blurring) that may affect PVAT identification. As shown in Fig. 2, this dilated area represented the perivascular region. Voxels corresponding to PVAT in the perivascular region that fell within the HU range of $[-190, -30]$ were identified (Nguyen et al., 2024). Biomarkers were calculated from these PVAT voxels: volume, mean and std. dev. (SD) of CT attenuation, and fat fraction (ratio of the number of PVAT voxels to the number of perivascular region voxels). To account for variations in scanners, acquisition, and disease characteristics, the PVAT CT attenuation was normalized by the average vessel lumen attenuation (Chatterjee et al., 2022).

### 2.5. Statistical Analysis

Dice similarity coefficient (DSC) and Hausdorff Distance (HD) error was used to assess the portal vein segmentation on 25 UW CT volumes. Several uni- and multi-variate logistic regression models were built (LR, scikit-learn package, Python v3.12) to diagnose the various fibrosis stages (advanced fibrosis group included cirrhosis patients). The UW dataset was

divided into training (80%, n = 385 patients) and testing (20%, n = 95 patients) subsets. The test set had 19 randomly sampled patients from each fibrosis stage.

Regression analysis was done following the same approach as prior work (Tallam et al., 2022). Logistic regression models were built with various combinations of the serum (APRI and FIB-4 scores) and automated CT biomarkers as features. A deep classifier could not be trained with the limited sample size in this work, which motivated the choice of logistic regression models. For univariate analysis, multinomial logistic regression models were created for each of the 33 features separately. For the multivariate analysis, a stepwise approach determined the optimal set of explanatory features. Adjustment for multiple comparisons was not done. Instead, a general rule-of-thumb $p$-value threshold of 0.0015 (0.05/33) was used to automatically identify the best features for final multivariate modeling.

For the multivariate model with the best features, a further sub-analysis was conducted. Three other models were created from the best model: (1) without both serum and PVAT features, (2) with serum but without PVAT features, and (3) without serum but with PVAT features, respectively. To compare against the step-wise multivariate approach above and determine the best methodology for optimal feature selection, another multivariate logistic regression model was trained with all the biomarkers and the feature importance was automatically determined for this model. The features with absolute log-odds ratios (logistic regression coefficients) greater than 0.05 were automatically identified. This resulted in 5 features for both cirrhosis and advanced fibrosis being chosen. A new multivariate model, denoted by "Top-5 features model", was trained with these features.

Diagnostic performance was assessed with AUC (and 95% confidence intervals), sensitivity, and specificity. A model with an AUC above 0.6 was considered effective. A bootstrapped DeLong test (MLstatkit package, Python v.3.12) compared the ROC curves from two models. A $p$-value $< 0.05$ was considered statistically significant.

## 3. Results

### 3.1. Portal Vein Segmentation Performance

Table 1 and Fig. 2 show the 3D nnU-Net segmentation performance for delineating the portal vein and its branches (left and right). The model segmented the main portal vein branch with a DSC of 85.8 ± 10.2 and HD error of 3.0 ± 2.4 mm. Compared to the main branch, higher HD errors were seen for the left branch (7.1 ± 6.9 mm) and right branch (7.9 ± 8.7 mm), respectively. Segmentation of the full portal vein (combination of main and all branches) was satisfactory with a DSC of 86.4 ± 9.6 and HD error of 6.9 ± 7.1 mm. It took nnU-Net ∼3 minutes to segment the portal vein and its branches in a CT volume.

### 3.2. Univariate Results

Table 2 shows the results from the univariate models. For cirrhosis, FIB-4 score obtained an AUC of 88.5% and specificity of 89.5%, but the sensitivity was lower at 73.7%. Among the CT features, spleen volume yielded the highest AUC of 89.2% and specificity of 94.7%. The SD of normalized CT attenuation for the right portal vein was the best PVAT feature with an AUC of 79.6%. For advanced fibrosis, FIB-4 score and spleen volume achieved the

highest AUCs among the serum and CT imaging features, respectively. However, AUC for the SD of normalized CT attenuation for the main portal vein was the highest at 67.1%.

Table 1: Results of portal vein segmentation by 3D nnU-Net on the UW dataset. PV: Portal Vein. DSC: Dice Score. HD: Hausdorff Distance error in mm.

| Structure | DSC (%) ↑ | HD (mm) ↓ |
|---|---|---|
| Main PV | 85.8 ± 10.2 (IQR: 82.8, 92.3) | 3.0 ± 2.4 (IQR: 1.6, 3.1) |
| Left PV | 81.4 ± 12.8 (IQR: 76.8, 92.0) | 7.1 ± 6.9 (IQR: 3.3, 8.4) |
| Right PV | 82.5 ± 14.6 (IQR: 76.7, 94.6) | 7.9 ± 8.7 (IQR: 0.8, 9.9) |
| Full PV | 86.4 ± 9.6 (IQR: 80.3, 92.9) | 6.9 ± 7.1 (IQR: 2.4, 9.0) |

Table 2: Results from the uni- and multi-variate logistic regression models with 95% confidence intervals. Blood serum scores (FIB-4 and APRI), Liver Surface Nodularity (LSN), Portal Vein (PV), Perivascular Adipose Tissue (PVAT), standard deviation (SD) of normalized (Norm) CT attenuation (Att). Best values are bolded. "-" indicates unknown value.

| | Advanced Fibrosis | | | Cirrhosis | | |
|---|---|---|---|---|---|---|
| | AUC | Sensitivity | Specificity | AUC | Sensitivity | Specificity |
| FIB4 | **83.2 (74.1, 91.6)** | 68.4 (52.6, 82.9) | 86.0 (75.4, 94.6) | 88.5 (78.0, 96.7) | 73.7 (50.0, 91.7) | 89.5 (82.2, 95.8) |
| APRI | 73.8 (62.0, 84.1) | 52.6 (35.5, 68.4) | **87.7 (78.0, 94.9)** | 82.3 (68.9, 93.0) | 73.7 (53.3, 93.3) | 84.2 (75.6, 92.0) |
| Spleen Volume | 82.0 (71.7, 90.9) | 71.1 (56.5, 84.6) | 82.5 (71.2, 91.4) | **89.2 (77.1, 98.3)** | 73.7 (53.3, 93.4) | **94.7 (88.9, 98.8)** |
| LSN | 76.3 (64.8, 86.7) | 65.8 (50.0, 79.5) | 80.7 (70.2, 91.2) | 88.1 (79.9, 95.1) | 78.9 (57.7, 94.7) | 88.2 (80.3, 94.9) |
| Main PV PVAT SD Norm. Att. | 67.1 (55.6, 77.4) | 50.0 (34.9, 65.8) | 77.2 (65.5, 88.0) | 70.3 (56.3, 83.1) | 63.2 (41.7, 84.6) | 68.4 (57.5, 78.5) |
| Right PV PVAT SD Norm. Att. | 64.5 (53.1, 74.6) | **78.9 (65.7, 91.2)** | 49.1 (35.7, 61.7) | 79.6 (69.0, 88.6) | **94.7 (83.3, 100.0)** | 53.9 (42.3, 64.6) |
| Lee et al. (2022) | 80 (69, 91) | - | - | 94 (89, 99) | - | - |
| Lewis et al. (2024) | 83.9 | 93.3 | 66.7 | 92.7 | 86.7 | 93.3 |
| Baseline | 84.8 (75.9, 92.2) | 68.4 (53.3, 83.3) | 87.7 (77.8, 95.3) | 91.6 (82.0, 98.5) | 73.7 (53.3, 93.4) | **97.4 (93.2, 100.0)** |
| Baseline & Serum | 88.5 (81.0, 94.8) | **86.8 (75.0, 97.1)** | 73.7 (61.4, 84.6) | 91.8 (83.5, 98.4) | 78.9 (57.9, 95.9) | 96.1 (91.0, 100.0) |
| Baseline & PVAT | 83.2 (74.3, 90.8) | 55.3 (37.8, 71.0) | **94.7 (88.6, 100.0)** | 92.9 (86.5, 98.1) | 78.9 (58.8, 95.2) | 90.8 (84.0, 96.3) |
| Baseline & Serum & PVAT | **88.7 (81.2, 94.7)** | 84.2 (72.5, 94.6) | 73.7 (62.3, 85.4) | **93.3 (86.5, 98.5)** | 78.9 (57.9, 94.7) | 93.4 (87.3, 98.7) |
| Top-5 features model | 84.5 (75.5, 92.8) | 81.6 (69.4, 92.3) | 80.7 (69.6, 90.6) | 90.9 (83.4, 96.9) | **84.2 (66.7, 100.0)** | 80.3 (71.0, 88.3) |

### 3.3. Multivariate Results

Table 2 also provides the multivariate modeling results. Figure 3 shows ROC curves of the different multi-variate models for advanced fibrosis and cirrhosis. Inference using the multivariate models took ∼2 minutes to perform for staging fibrosis.

**Cirrhosis:** Out of 33 features, 16 features including serum and PVAT were indicative for cirrhosis. They included: (1) LSVR, (2) LSN, (3) spleen volume, (4-10) volume proportions of Couinaud segments except segment 7, (11) FIB4, (12) APRI, (13-15) normalized SD CT attenuation for the main, left, and right portal veins, and (16) fat volume at the right portal vein. The "baseline" model had the best CT imaging features for predicting cirrhosis, excluding serum and PVAT scores, and achieved a 91.6% AUC and 97.4% specificity. Adding both serum scores slightly improved the AUC by 0.2%, but specificity dropped by 1%. Adding PVAT biomarkers to baseline increased AUC to 92.9%. However, the specificity fell to 90.8%. Adding both serum and PVAT features improved the baseline AUC to 93.3% with a specificity of 93.4% and sensitivity of 78.9%. All multivariate models were significantly different ($p < .05$) from APRI score, and normalized mean and SD CT attenuation for main and right portal vein. No statistical difference was seen between the multivariate models.

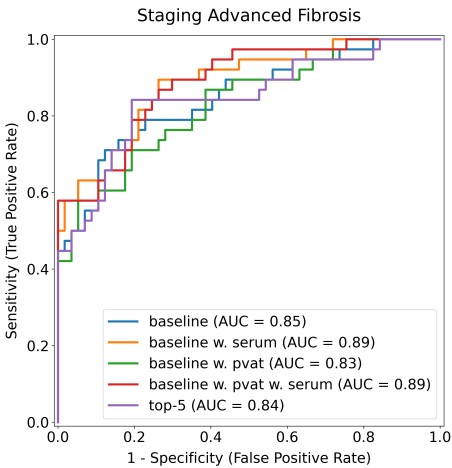 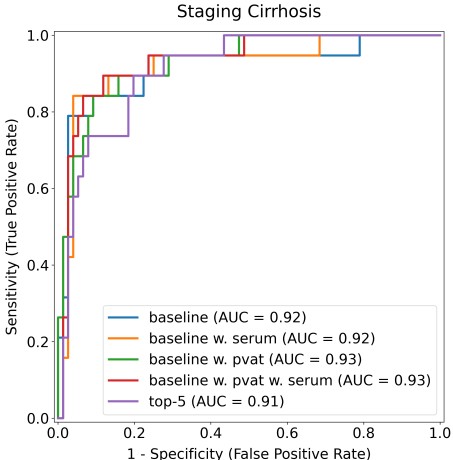

Figure 3: ROC curves for staging advanced fibrosis (left) and cirrhosis (right) with multivariate logistic regression models.

The top-5 features multivariate model contained the following: FIB4, APRI, left portal vein PVAT mean and SD attenuation, and LSN. Compared to the other multivariate models, the top-5 features model obtained the lowest AUC (90.9%) and specificity (80.3%) for diagnosing cirrhosis, but had the highest sensitivity (84.2%) due to the misclassification trade-off (false positives vs. false negatives). This model was significantly different ($p < .05$) from all univariate features except for spleen volume and FIB4. However, the top-5 model showed no difference between the examined multivariate models.

**Advanced Fibrosis:** Out of 33 features, the same 16 features indicative for cirrhosis were also indicative for advanced fibrosis. The baseline model with only CT features, excluding PVAT and serum, attained 84.8% AUC and 87.7% specificity. Addition of both serum scores improved the AUC by ∼4%, but the specificity dropped to 73.7%. Addition of PVAT features reduced AUC to 83.2%, but specificity was the highest at 94.7%. Finally, adding both PVAT and serum features to the baseline increased AUC to 88.7% with a sensitivity of 84.2% and specificity of 73.7%. This multivariate model was significantly different ($p < .05$) from all univariate models except FIB4 and spleen volume.

The top-5 features model contained the same 5 features as the model for cirrhosis. Compared to the other multivariate models, the top-5 model achieved an AUC of 84.5%, specificity of 81.6%, and sensitivity of 80.7%. This model was significantly different ($p < .05$) from all univariate models except for LSVR, spleen volume, LSN, FIB4, and mean and standard attenuation of the right portal vein. Again, statistical testing showed no difference between the examined multivariate models, including the top-5 model.

## 4. Discussion

The main contributions in this work included: (1) automated fine-grained segmentations of the portal vein and its branches (left and right), and (2) use of a novel PVAT CT biomarker for staging fibrosis. A DSC score of 86.4 ± 9.6 was obtained for the segmentation of the portal vein. Our results are similar to those obtained by prior approaches (Li et al., 2024;

Ibragimov et al., 2017), wherein DSC scores of 70% - 89% were achieved. As a result, the portal vein segmentation by the 3D nnU-Net model was deemed satisfactory for clinical use.

With the serum and PVAT biomarkers, multivariate models were trained to achieve AUCs of 91% - 93.5% and 83% - 89% for cirrhosis and advanced fibrosis, respectively. In prior literature by Lee et al. (2022) on the same UW dataset (see Table 2), AUCs of 94% and 80% were obtained for diagnosing cirrhosis and advanced fibrosis, respectively. Similarly, Lewis et al. (2024) obtained AUCs of 92.7% and 83.9% for cirrhosis and advanced fibrosis, respectively. Both used the same testing dataset and explainable CT biomarkers for staging hepatic fibrosis, such as spleen volume, LSVR, and LSN. Choi et al. (2018) used neural networks to directly stage fibrosis with AUCs of 0.97 and 0.95 achieved for advanced fibrosis and cirrhosis, respectively. However, this approach required a large proprietary dataset of >7000 patients, rendering it challenging to replicate due to a lack of publicly available datasets with a large number of patients and confirmed fibrosis stages.

A major strength of this work was that the explainable CT imaging biomarkers were the same for both advanced fibrosis and cirrhosis. This observation indicated that these features are relatively stable across different fibrosis stages. It is also remarkable that the top-5 multivariate feature model (automatically selected based on log-odds scores) also selected the same stable CT biomarkers for both cirrhosis and advanced fibrosis. Despite the lack of statistical significance, this corroborates the benefit of explainable CT imaging features, especially for PVAT, in capturing physiological processes for staging fibrosis.

Limitations of this work are acknowledged. First, the PVAT biomarkers were not very predictive on their own for advanced fibrosis or cirrhosis. This may be attributed to the underlying patient pathophysiology. For example, upon a qualitative analysis, eight patients amongst the 19 cirrhotic patients in the UW test dataset had ascites (fluid buildup in the abdomen). As shown in Fig. 2, ascites masked the visceral fat in the periportal region and appeared brighter than the CT attenuation range of [-190, -30] for fat. Fewer PVAT voxels were identified as a result. The predictive value of PVAT biomarkers increased when combined with other features, such as serum or other CT imaging biomarkers (e.g., spleen volume). Next, periportal space widening may be more pronounced in certain patients in contrast to others, such as lean individuals with lower BMI (Ludwig et al., 2021). As there is scant research on automated techniques to measure periportal fat (Song et al., 2024; Ludwig et al., 2021; Ito et al., 2000), our findings hold value for furthering research into hepatic PVAT. Second, the dilation kernel can also be specified in mm (instead of voxels) based on the spacing of each CT volume in a dataset. The effect of an asymmetrical dilation kernel on PVAT estimation has not been evaluated in this work. Third, PVAT biomarkers were not extracted from the hepatic arteries and hepatic veins in this work. The diagnostic performance may improve with their addition. Fourth, the automated approach was not validated on non-contrast CT. Lastly, the sample size used in this work was small (19 patients from each fibrosis stage). External testing is necessary to further clarify our findings. This is the subject of future work.

In summary, the non-invasive and explainable CT imaging-based biomarkers of PVAT derived in this work were useful for staging advanced fibrosis and cirrhosis. Predictive CT biomarkers were the same for both categories, indicating the stability of these features for staging fibrosis. The current approach shows promise for population-based studies of chronic metabolic diseases and opportunistic screening.

## Acknowledgments

This work was supported by the Intramural Research Program of the NIH Clinical Center (project number 1Z01 CL040004). The research used the high-performance computing facilities of the NIH Biowulf cluster.

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
