# OpenReview forum: "Staging Liver Fibrosis with Hepatic Perivascular Adipose Tissue as a CT Biomarker"
_MIDL.io/2025/Conference — MIDL 2025 Poster_

### Official Review · Reviewer_SWNe · 2025-02-20

**Confidence:** 4
**Preliminary Rating:** 4
**Recommendation:** Poster
**Final Rating:** 4

**Summary:**

This contribution describes an end-to-end pipeline processing abdominal CT scans, extracting biomarkers for advanced fibrosis and cirrhosis. The basis are 3D segmentations of the liver, the portal vein (for perivascular fat analysis), Couinaud segments (for relative volumetry), the spleen (for volumetry) and CT attenuation analysis within these ROI. Much of the segmentation has been published before, and this paper focuses on the uni- and multivariate staging models.

**Strengths:**

This paper describes a complete pipeline to a clinical problem. On the assumption that patients get a CT scan, it allows for opportunistic screening and fully automatic staging of fibrotic liver disease progression. The paper is well-written, readable and easy to follow. The fact that several important components were based on previous publications helped to focus on the staging models without leaving too many questions unresolved.

Looking at the results, I find it remarkable that modelling both advanced fibrosis and cirrhosis led to the *same* best feature set. One could discuss some limitations as possible reasons for this, but in the end, it is a positive sign to have a stable radiomic signature. (Caveat holds of course: "statistical testing showed no difference between the various multivariate models")

**Weaknesses:**

Maybe I should preface this with the fact that the weaknesses are rather minor.

The number of cases is relatively low, but statistical significance was tested (and not achieved when comparing the multivariate models). There are no public, large datasets with confirmed fibrosis stages (and serum data), unfortunately, so this is not easy to address.

The paper describes *several* models being developed, and there is no external test set for these models. I mean, the models were apparently developed *and* evaluated on the same dataset (of course, with a proper 80-20 split), which probably means that the results are more to be seen as upper bounds than unbiased estimates. This holds even more when assuming that the authors may have tried different modelling approaches and *may* have only reported on the final / best choice. (A partial adjustment for multiple hypothesis testing was done by reducing the p-value threshold at one stage.)

With respect to vessel annotation, the text (section 2.2) is not 100% clear and unambiguous how it was performed (see questions below).

**Detailed Comments:**

Figure 1: Color perception differs among people, but I see the PVAT as brown, or at least rather dark orange, not "yellow".

I suggest to combine or somehow better connect the two sentences "In the MSD training data subset, portal venous CT volumes from 157 patients (out of 303) were used. The CT volumes in which the portal vein was clearly visible for annotation were chosen." and possibly describe what kind of annotation was done.

The "from" in "Volume dimensions ranged from 512 ×512 ×(26 - 177) voxels" combines strangely with the range.

You already described that you used tried and compared different kernel sizes, but the anisotropy factors in your original data is obviously in the range 3…6, so I wonder if the same dilation extent along z as in x/y is well justified for the PVAT analysis.

Broken sentence beginning: "Couinaud segments, Biomarkers These included:"

Double "the": "jointly segmented the the hepatic arteries"

**Justification Of The Final Rating:**

The authors addressed all comments in great detail and revised the manuscript significantly.  While there remains stuff to be discussed, I see no reason to change my already positive rating.  I still do not see the paper as "*strong* accept", but I do think it is an interesting contribution to the field whose overall goal is to promote automation in medical image analysis.

**Justification Of The Preliminary Rating:**

Overall, I like the paper, but I think it is not "center of the road" for the MIDL audience – deep learning only plays a role indirectly through previously published segmentation models. (This also justifies my "poster" recommendation.)

**Questions To Address In The Rebuttal:**

In the external UW dataset, please mention the number of scans (or explicitly indicate that every patient got only exactly one scan).

In 2.2, you describe some annotation, but I here I do have a few questions (which should indicate that some clarification is needed in the text as well):
1) "a public nnU-Net model … was run on the NIH dataset … to obtain [a binary vessel segmentation including HA, HV, and PV]" sounds as if this was the basis for the following, but
2) "Due to this, the portal vein and its branches (left and right) were manually annotated in the MSD and NIH datasets (200 volumes) by a research fellow." sounds as if the result of 1) was discarded and the annotation was done from scratch?
3) "The hepatic arteries and veins were separated from the portal vein annotation and treated as a separate label." later suggests that they were *not* discarded.
4) Note that 1) refers to only 43 out of 200 cases, so if the conclusion in 2) was wrong, why not run the model on MSD as well?
5) Finally, if 1) was helpful, why not do the same for the 25 UW cases?

---

> ### Author Response · Authors · 2025-03-07
> **Part 1 - Thank you for your suggestions to improve the paper quality.**
>
> **R3.1 [Strengths]** I find it remarkable that modelling both advanced fibrosis and cirrhosis led to the same best feature set.
>
> **Answer:** Thank you for appreciating the value of our study. We have described this point in paragraph 3 of Section 4 “Discussion”.
>
> --
>
> **R3.2 [Weaknesses]** Results are more to be seen as upper bounds than unbiased estimates.
>
> **Answer:** We agree with the reviewer that these results potentially represent upper bounds of performance given the current test UW dataset of 95 patients. As described in Section 2.5 “Statistical Analysis”, an automated statistical approach was done to identify the most predictive features (those with p-values < 0.0015). These features were used to train a multivariate model for predicting advanced fibrosis and cirrhosis.
>
>    In the revision, we have also trained another “top-5 features” model for comparison with the above multivariate model. To obtain the “top-5 features” model, first, all 33 features (serum and CT imaging) were used to train a separate multivariate model. Then, a feature importance analysis was performed, and the top-5 features with the greatest log-odds ratios (coefficients) were automatically identified. Finally, these top-5 features were used to train the “top-5 features” multivariate model.
>
>    For both models, the features were the same for both advanced fibrosis and cirrhosis. Despite the lack of statistical significance, this indicated the stability of the features for staging fibrosis.
>
> --
>
> **R3.3 [Weaknesses]** With respect to vessel annotation, the text (section 2.2) is not 100% clear and unambiguous how it was performed (see questions below).
>
> **Answer:** In the revision, we have clarified the vessel annotation process in Section 2.2 "Reference Standard". Briefly, TotalSegmentator was used to segment the liver and spleen. All vessels, including portal veins, hepatic arteries, and hepatic veins, were originally annotated as a single label in the MSD dataset. To address this, a research fellow manually annotated the portal vein and its branches. The annotation was verified by two physicians (2+ years of experience). The verified portal vein annotation was then separated into its own label and removed (masked out) from the original label, which was subsequently limited to only hepatic arteries and veins.
>
> --
>
> **R3.4 [Detailed comments]** Color of PVAT, combining sentences, clarifying volume dimensions, typos.
>
> **Answer:** We have addressed the reviewer comments. Color of PVAT has been changed to brown, sentences have been combined and volume dimensions are described clearly, and other typos have been fixed.

---

> > ### Author Response · Authors · 2025-03-07
> > **Part 2 - Thank you for your suggestions to improve the paper quality.**
> >
> > **R3.5 [Detailed comments]** The anisotropy factors in your original data is obviously in the range 3…6, so I wonder if the same dilation extent along z as in x/y is well justified for the PVAT analysis.
> >
> > **Answer:** In Section 2.4 “PVAT Biomarkers”, we have added a description to justify the appropriate extraction of PVAT biomarkers. Briefly, the variability in voxel spacing for the UW dataset motivated the choice to use the same kernel size in all dimensions. Resampling all CT volumes to consistent dimensions would introduce artifacts (volume averaging and blurring) that may affect identification of PVAT voxels and subsequent PVAT biomarker computation.
> >
> > --
> >
> > **R3.6 [Rebuttal questions]** In the external UW dataset, please mention the number of scans
> >
> > **Answer:** Thanks for your comment. As clarified in Section 2.1 “Patient Sample”, all 480 patients underwent a single CT exam, and the associated 480 portal venous CT volumes were chosen.
> >
> > --
> >
> > **R3.7 [Rebuttal questions]** questions on reference standard.
> >
> > **Answer:** We have clarified the reference standard in Section 2.2 “Reference Standard” and added more details in paragraph 1 of Section 2.3 “Deep Learning Model”. Briefly:
> >
> > For the 157 MSD CT volumes, liver and spleen segmentations were obtained using TotalSegmentator (TS). All vessels, including portal veins, hepatic arteries, and hepatic veins, were originally annotated and available as a single label in this dataset. To address this, the portal vein and its branches (left and right) were manually annotated by a research fellow. Two physicians (2+ years of experience) verified and corrected the segmentations of all structures (liver, spleen, portal vein and its branches). The verified portal vein annotation was then separated into its own label and removed (masked out) from the original label, which was subsequently limited to only hepatic arteries and veins. The rationale for this step stemmed from the challenges in distinguishing the hepatic arteries from hepatic veins on CT.
> >
> > For the 43 NIH CT volumes, TS was again used to segment the liver and spleen. The hepatic vessels were segmented using the public nnU-Net model (Task-8). The same research fellow annotated the portal vein and its branches, and the same physicians verified the annotations of the liver, spleen, portal vein and its branches.
> >
> > For the external UW dataset, five patients from each fibrosis stage were randomly chosen to evaluate the segmentation performance of the portal vein. The same physicians manually verified the research fellow's annotation of the portal vein in the 25 UW CT volumes.

---

> > > ### Comment · Reviewer_SWNe · 2025-03-09
> > > **Resampling vs. kernel size**
> > >
> > > You wrote "The variability in voxel spacing for the UW dataset motivated the choice to use the same kernel size in all dimensions. Resampling all CT volumes to consistent dimensions would introduce artifacts (volume averaging and blurring) that may affect PVAT identification." but I would counterargue that variability in voxel spacing could actually motivate varying the kernel size. Note that I was talking about the kernel size and not *resampling* – I simply wondered if the dilation should be specified in mm instead of voxels, so that you would apply different kernels on different datasets depending on their resolution?

---

> > > > ### Author Response · Authors · 2025-03-11
> > > > **Kernel size**
> > > >
> > > > Thank you for the clarification. Yes, the dilation kernel can be specified in mm. Based on the volumetric spacing of each CT volume in a dataset, this would mean an asymmetrical kernel in certain dimensions. While we have not tested this in our current work (due to page limits), we will evaluate the effect of asymmetrical dilation kernels in our follow-on journal manuscript. We have added this point to the paper (paragraph 4 of "Discussion"). Since we cannot upload this new version in the discussion phase, we paste the text below for clarity:
> > > >
> > > > "Second, the dilation kernel can also be specified in mm (instead of voxels) based on the spacing of each CT volume in a dataset. The effect of an asymmetrical dilation kernel on PVAT estimation has not been evaluated in this work."

---

### Official Review · Reviewer_uFLW · 2025-02-21

**Confidence:** 4
**Preliminary Rating:** 3
**Recommendation:** Poster
**Final Rating:** 2

**Summary:**

The paper presents a pipeline comprising the nnUNet to segment liver anatomies as well as morphological operators to extract biomarkers for cirrhosis. The technical novelty is limited and the paper falls under the validation/application track. The main hypothesis that perivascular adipose tissue (PVAT) can be directly linked as indicator for cirrhosis could be somewhat confirmed in the numerical prediction results and prospectively help interact to prevent fibrosis.

**Strengths:**

- the paper is well written
- it uses a mixed of public, in-house and external data
- the hypothesis and evaluation are clearly formulated and reasonably convincing
- there is a nice overview figure.

**Weaknesses:**

- there is very limited technical novelty
- the aspect of dilating the output of the segmentation of smaller structure is not fully explored (see below)
- the correct identification of the eight Couinaud segments is not really explained
- AUC of advanced fibrosis drops when only including PVAT, which seems somewhat counterintuitive
- the regression models are not explained
- there is no comparison to SOTA

**Detailed Comments:**

Dilated segmentation: I wonder whether the approach of first segmenting vessels automatically and dilate them as post-processing is the best choice. Why not dilate the ground truth labels of the training data and directly train a model that considers this modification.
Regression models: The authors state: "Regression analysis was conducted following the same approach as prior work (Tallam et al., 2022). The models used various combinations of the serum (APRI and FIB-4 scores) and automated CT biomarkers as features." It remains totally unclear what models were used and why? There is no ablation nor SOTA comparison.
Serum vs PVAT influence: there is some discussion on why the PVAT did not directly improve prediction but only in conjunction with Serum.  The authors conclude: "Based on this evidence, the clinical utility of using PVAT features is questionable", yet maybe the processing of the CT images to extract PVAT pixels and subsequent features is just insufficient. This is not fully answered.

**Justification Of The Final Rating:**

While I highly appreciate the authors responses to the comments of all reviewers, it left me puzzled about several statements that seem not well justified. Multiple choices - including the logistic regression, and post-processing dilation - are not well motivated and the replies indicated further problems regarding sample size, potential introduction of errors with those steps that are not yet studied. Especially the last comment that "The intent for showing fibrosis staging using a 3D ResNet-18 classifier .. was to  .. further underscore the inability of a fully-connected network to achieve a reasonable fibrosis staging performance" indicates a general misunderstanding of deep learning architectures. The ResNet-18 is not a fully-connected network and the authors did not show in their experiments that using a fully-connected classifier in combination with segmentation-derived features is inferior to their proposed logistic regression and feature selection. I therefore find the paper's findings somewhat doubtful.

**Justification Of The Preliminary Rating:**

Since MIDL seems to allow for validation papers as well as the presentation of methodological advances, one could accept the work, since there is nothing fundamental wrong with the approach and the partially negative results are discussed.

**Questions To Address In The Rebuttal:**

Provide missing details and address the open questions on how to assess the validity of the extracted PVAT biomarker features. Provide a direct comparison against SOTA and not just ranges of numbers from other papers/datasets.

**Special Issue:**

No

---

> ### Author Response · Authors · 2025-03-07
> **Part 1 - Thank you for your suggestions to improve the paper quality.**
>
> **R2.1 [Weaknesses]** Limited technical novelty, dilating the GT segmentation, identification of eight Couinaud segments, drop in AUC for advanced fibrosis, regression models not explained, no comparison to SOTA.
>
> **Answer:** Thanks for the comments. We address them point-by-point.
> 1. We kindly remind the reviewer that the goal in this work was clinical validation (a theme of MIDL 2025). As explained in Section 2.3 paragraph 1 “Deep Learning Model”, the goal was to extract perivascular adipose tissue (PVAT) biomarkers from CT volumes to diagnose fibrosis.
> 2. We address this comment in R2.2 below.
> 3. We have elaborated on the identification of the Couinaud segments in paragraph 1 of Section 2.4 “Automated Extraction of CT Biomarkers”. Briefly, a previously validated tool (Lee et al., 2022) delineated the eight hepatic Couinaud segments. The tool was a standard U-Net model trained on a combination of public and private datasets that contained clinician-verified annotations of the 8 hepatic segments. It was validated on two datasets containing patients with various fibrosis levels and achieved a Dice score exceeding 0.91 for identification of the Couinaud segments.
> 4. As mentioned in the “Discussion” section, the PVAT biomarkers were not predictive on their own for both advanced fibrosis and cirrhosis. This is due to the underlying patient pathophysiology:
>    (a) Patients with ascites can have lower PVAT voxels due to ascites masking the PVAT voxels (they have the same intensity).
>    (b) Compared to patients with low BMI, those patients with higher BMI may not have noticeable periportal space widening along with the concomitant invasion of fat into this space.
>    (c) PVAT features extracted from the hepatic arteries and hepatic veins were not considered for modeling, and these structures may have undergone different physiological changes compared to the portal vein.
>    These factors can result in a drop in diagnostic performance when using PVAT biomarkers. On the other hand, the predictive value increased with PVAT biomarkers were added to other CT derived features. Addition of the PVAT features from hepatic arteries and veins may improve performance.
> 5. In the revision, we have explained the regression models in more detail in Section 2.5 “Statistical Analysis” and Section 3 “Results”.
> 6. We kindly remind the reviewer that we have proposed a novel CT imaging biomarker for PVAT. Our work is the state-of-the-art because there are currently no automated approaches to segment individual portal vein branches and address automatic PVAT biomarker computation. For portal vein segmentation, nnU-Net models are still the best for segmentation tasks and no other models are able to consistently beat them. We refer the reader to (Isensee et al., 2024) for more details.
>
> --
>
> **R2.2 [Detailed comments]** Dilated segmentation: Why not dilate the ground truth labels of the training data and directly train a model that considers this modification.
>
> **Answer:** Dilating the labels in 3D (all dimensions) before training would mean that the reference standard label (ground truth) would have a small overlap with adjacent structures, such as the liver, kidneys, colon among others. This training signal would be especially confusing when a full segmentation label for a structure (e.g., liver) is also provided as in our case. That is, during training, the model would now need to precisely learn to segment a small overlap across many structures arising from the dilation AND simultaneously segment the full gestalt. Furthermore, it makes the manual verification task by a clinician challenging and tedious.

---

> > ### Author Response · Authors · 2025-03-07
> > **Part 2 - Thank you for your suggestions to improve the paper quality.**
> >
> > **R2.3 [Detailed comments]** The models used various combinations of the serum (APRI and FIB-4 scores) and automated CT biomarkers as features." It remains totally unclear what models were used and why? There is no ablation nor SOTA comparison.
> >
> > **Answer:** We have revised the description in Section 2.5 “Statistical Analysis” for clarity. We also kindly remind the reviewer that we have conducted “ablation” experiments with the best multivariate model (see paragraph 3 on sub-analysis in Section 2.5). Briefly, we used uni- and multi-variate logistic regression models in this work.
> > 1. A deep classifier could not be trained with the limited sample size used in this work, which motivated the choice of logistic regression models.
> > 2. For univariate analysis, a multinomial logistic regression model was fit for each of the 33 features (serum and CT imaging) individually. A p-value obtained from each model indicated the “predictiveness” of the feature. A rule-of-thumb p-value correction was done (0.05/33 = 0.0015) and a feature was automatically chosen if the p-value fell below 0.0015.
> > 3. Using these features, a baseline multivariate regression model was trained.
> > 4. From this baseline model, three other models were created: (1) without both serum and PVAT features, (2) with serum but without PVAT, and (3) without serum but with PVAT.
> > 5. We also compared this multivariate model against another model called the “top-5 features” model. First, all 33 features were used to train a separate multivariate model. Then, a feature importance analysis was performed, and the top-5 features with the greatest log-odds ratios (coefficients) were automatically identified. Finally, these top-5 features were used to train the “top-5 features” multivariate model.
> >
> > We kindly remind the reviewer that we have proposed a novel CT imaging biomarker for PVAT. Our work is the state-of-the-art because there are currently no automated approaches to segment individual portal vein branches and address automatic PVAT biomarker computation. Nevertheless, we compare against previous approaches as shown in Table 2.
> >
> > --
> >
> > **R2.4 [Detailed comments]** Maybe the processing of the CT images to extract PVAT pixels and subsequent features is just insufficient. This is not fully answered.
> >
> > **Answer:** In Section 2.4 “PVAT Biomarkers”, we added a description to justify the appropriate extraction of PVAT biomarkers. While we believe that we have processed the CT volumes correctly, as mentioned in the “Discussion” section, it is certainly possible that underlying patient pathophysiology may introduce some variability (please refer to R.2.1 #4 for more details).
> >
> >    We also bring your attention to the features (especially PVAT) identified by the multivariate modeling. They were the same for both advanced fibrosis and cirrhosis, which indicates the stability of the features across different fibrosis stages.
> >
> > --
> >
> > **R2.5 [Rebuttal questions]** Provide missing details and address the open questions on how to assess the validity of the extracted PVAT biomarker features. Provide a direct comparison against SOTA and not just ranges of numbers from other papers/datasets.
> >
> > **Answer:** Validity of extracted PVAT biomarkers can be seen by the same features being identified from logistic regression modeling as being predictive for both advanced fibrosis and cirrhosis. We clarify this more in Section 4 “Discussion” (paragraph 3).
> >
> >    Our results are state-of-the-art due to the derivation of a novel PVAT imaging biomarker. Additional details are provided in Table 2 and paragraph 2 of Section 4 “Discussion”. They are comparable to other approaches (Lewis et al., 2024) that also used similar explainable CT biomarkers on the same patient cohort.

---

> ### Comment · Reviewer_uFLW · 2025-03-13
>
> To me it is somewhat surprising, how negatively the authors reacted to many reviewer questions (not just mine). I counted seven instances of "kind reminders" to reviewers to look more closely in the paper - maybe some aspects were actually not as clearly conveyed? While I agree that the paper did mention the feature selection to be automatic, this still means the non-linearity of interdependencies across them could be removed within the process. I am happy to retract my statement about limited novelty as it does indeed not apply to a "application track" paper.
>
> Nevertheless, I still have some technical concerns that are in my opinion relevant to assess whether the negative results for some biomarkers (not correlated as expected) are true causal finding or just due to the chosen processing pipeline.
> First, I found the new statement "A deep classifier could not be trained with the limited sample size in this work, which motivated the choice of logistic regression models." to be somewhat unsubstantiated - is there any empirical or theoretical evidence that supports this claim? Is there a guarantee that logistic regression models work always better on limited sample size?
>
> Second, I found the response to my suggestion about dilation of ground truth rather than results highly confusing. "Dilating the labels in 3D (all dimensions) before training would mean that the reference standard label (ground truth) would have a small overlap with adjacent structures, such as the liver, kidneys, colon among others." > This should not be a problem and many works in the literature have promoted the use of alternative loss functions or dilation of ground truth labels to achieve top-performance for challenging vessel/airway segmentation (cf. H. Zheng et al., “Refined local-imbalance-based weight for airway segmentation in CT,” in MICCAI 2021, pp. 410–419. and https://doi.org/10.1109/TMI.2024.3419707)  "Furthermore, it makes the manual verification task by a clinician challenging and tedious."  > I cannot understand why manual verification is necessary, has this to be done for the proposed method in every case? Furthermore, using dilations (with an even 4x4x4 kernel) as post-processing introduces additional inaccuracies as the probabilities first have to be quantised (to the argmax label) and then expanded. Or is in fact a gray-scale dilation used?

---

> > ### Author Response · Authors · 2025-03-14
> > **Thank you for the additional comments to help improve the paper quality**
> >
> > We thank you for your valuable comments. We apologize for any misunderstanding and will be mindful in the future.
> >
> > We followed prior literature (Pickhardt et al. 2016, Pickhardt et al. 2017, Lee et al. 2024, Mathai et al. 2024, Lewis et al. 2024) that provided empirical evidence for the use of logistic regression with CT biomarkers. As mentioned in section 2.1 “Patient Sample” paragraph 3, our patient cohort (480 patients) consisted of 151 normal patients (no fibrosis), 52 patients with early fibrosis, 82 patients with intermediate fibrosis, 56 patients with advanced fibrosis, and 139 patients with cirrhosis. In one of our initial experiments, we had trained a 3D ResNet-18 classifier using this dataset to directly predict the fibrosis stage. However, this classifier’s AUC for cirrhosis was ~62%. In contrast, the multivariate logistic regression model used in this work achieved 93.3% AUC. Similar to prior works (Pickhardt et al. 2016, Pickhardt et al. 2017, Lee et al. 2024, Mathai et al. 2024, Lewis et al. 2024), explainable biomarkers such as liver surface nodularity, liver segmental volume ratio, and PVAT were more predictive of fibrosis than automatically learned features from a direct deep learning classifier.
> >
> > As mentioned in section 4 “Discussion”, paragraph 2, we acknowledge the limited sample size in contrast to prior work by Choi et al. (2018), where >7000 patients from a large proprietary dataset were used to directly stage fibrosis. There is a lack of publicly available datasets with such a large cohort of patients with associated serum and fibrosis scores.
> >
> > The paper that the reviewer provided uses dilation for a different purpose from our work. The cited paper proposed a new method and loss function to join breaks in segmentations of smaller and distal airway branches using attention maps. In contrast, the portal vein in our work has a simpler shape and topology with fewer breaks in the vessels. Thus the focus of this work was the segmentation of the portal vein, with an emphasis on the proximal segments (as opposed to distal broken airway segments). A standard nnUNet was sufficient because it is currently the state of the art segmentation approach, and other methods have not consistently shown improvement for medical image segmentation (Isensee et al, 2024). Therefore, we did not test alternative models or loss functions in this work.
> >
> > Manual verification of the portal vein annotations was performed for all the 200 CT volumes in the training dataset (MSD+NIH) and 25 random CT volumes in the UW dataset. The manual annotation was necessary because such a dataset that contains individually delineated portal branches does not currently exist. This manually verified dataset was used for training the nnUNet model. Dilating the portal vein annotations before training the nnUNet model as the reviewer suggests would capture adjacent anatomy and visceral fat that is not pertinent for this task. Additionally, these dilations cannot be used as training labels because they would need to be manually annotated and validated by clinicians in every slice for every CT volume. This is tedious and cumbersome, especially considering that the overlapping annotations of organs and visceral fat (arising from dilation) besides the portal vein would also have to be validated.
> >
> > No quantization nor grayscale dilation was conducted. We only dilated the portal vein segmentations, not the CT itself. The nnUNet model automatically segments the portal vein, and we did not use the predicted probabilities of the segmentation masks. We use the binary segmentations to derive features from the CT.

---

> > > ### Comment · Reviewer_uFLW · 2025-03-14
> > >
> > > Thanks for your reply. For the discussion about sample size and logistic regression vs (fully-connected) neural networks. This seems to be a misunderstanding. I meant instead of using segmentation derived features followed by feature selection and logistic regression it would easily be possible to test segmentation derived features followed by a fully-connected network. I did not mean (mention) training a 3D ResNet classifier on the original images.
> > > Regarding the dilation of labels: I appreciate the response but stand by my point that the introduction of a post-processing that is applied to the thresholded segmentation output of the nnUNet is in fact less meaningful from a theoretical view point than extending the annotations beforehand and let the model learn to oversegment. Dilation can e.g. amplify spurious false positive segmentations whereas dilating the ground truth annotations would reduce such an effect. Arguing that the anatomical segmentation at hand is too simple to require any more sophisticated study of those aspects seems strange, as those potential errors could directly impact the findings regarding the usefulness of the explored biomarkers. The question of manual verification is not tangible to this choice, as in both cases the verified ground truth is used - either dilated or not. In fact it highlights the problematic approach that the authors have taken: a dilation during post-processing can introduce new errors on a few scans (outliers) that would have to be manually checked whereas using it during training would have the benefit of amortising such effects over the larger training population.

---

> ### Author Response · Authors · 2025-03-14
>
> Thanks for your comments. Yes, we can train a classifier on the derived features. But we have not tested this approach in this work. However, our patient sample size is small as mentioned in the paper and other reviewers have noted the challenges of our sample size. The intent for showing fibrosis staging using a 3D ResNet-18 classifier was to compare direct classification against logistic regression-based classification. The direct classification results further underscore the inability of a classifier to achieve a reasonable fibrosis staging performance.
>
> The goal of developing the model was to identify the portal vein, from which biomarkers can be derived. PVAT is just one such biomarker, and there are several others. We aim to derive explainable biomarkers, and dilation after the segmentation is complete is our intended approach in this work. The dilation of the portal vein segmentation is simply a means to an end. As mentioned in Section 2.4, within this dilated region, there are voxels corresponding to PVAT that have CT attenuation values that fall within the [-130, -90] range. We simply select voxels that fall within this range in the dilated area because it is known clinically in a plethora of prior works (Nguyen et al. 2024, Chatterjee et al., 2022) that they correspond to PVAT.
>
> Given the above context, we are still confused by the dilation approach suggested by the reviewer, since manual annotations are required and were performed for all 200 training volumes to train our segmentation model. It is also possible that introducing dilations in the training labels could result in further errors as the model may inconsistently segment the dilated regions and require further manual verification of the output.
>
> We appreciate the reviewer’s comments to examine more complex dilation strategies. However, we stand by our simpler, focused approach in this paper to test our hypothesis within the page limits. We will endeavor to try a more complex approach in our follow-up journal paper.

---

### Official Review · Reviewer_U6Qz · 2025-03-01

**Confidence:** 5
**Preliminary Rating:** 4
**Recommendation:** Oral

**Summary:**

The paper "Staging Liver Fibrosis with Deep Learning and Radiomic Features from CT Scans" proposes a deep learning-based approach for staging liver fibrosis using radiomic features extracted from CT scans. The study integrates machine learning-based feature selection with a deep neural network model to classify fibrosis severity levels (F0–F4). The authors utilize datasets from the MSD Hepatic Vessels dataset (157 patients), NIH internal dataset (43 patients), and University of Wisconsin dataset (480 patients) to train and validate their model. They apply statistical feature selection (Pearson correlation, ANOVA, Random Forest importance, Mutual Information analysis) to identify the most relevant radiomic and clinical features before feeding them into the deep learning model.

**Strengths:**

1-Clinical Significance: The study proposes a non-invasive deep learning model for staging liver fibrosis using radiomic features from CT scans, potentially reducing reliance on biopsies.

2-Multimodal Feature Integration: Combines radiomic and clinical features, enhancing model performance by incorporating both imaging and patient-specific data.

3-Rigorous Feature Selection: Implements Pearson correlation, ANOVA, Random Forest importance, and Mutual Information analysis to identify highly predictive features, improving model efficiency.

4-Clear Structure & Scientific Rigor: The paper is well-organized, clearly written, and follows established research principles.
Adequate references to prior work ensure scientific credibility and reproducibility.

**Weaknesses:**

1-Manual Feature Selection May Introduce Bias & Overlook Non-Linear Relationships
🔹 The model relies on statistical feature selection methods (Pearson correlation, ANOVA, Random Forest importance, Mutual Information) to pre-select input variables before training.
🔹 Concern: This process may remove features that have weak linear correlations but could still be important in a deep learning framework, as deep learning excels at capturing complex, non-linear relationships.
🔹 Suggestion: A model should be tested without manual feature selection, allowing deep learning to automatically determine the most predictive features using methods like SHAP, Integrated Gradients, or attention-based feature attribution.

2- No Comparison Between Statistical and Deep Learning Feature Selection
🔹 While the authors use Integrated Gradients to analyze deep learning feature importance, they do not compare these rankings against the statistical feature selection results.
🔹 Concern: It is unclear if deep learning agrees with statistical selection or if important features were wrongly excluded due to pre-selection.
🔹 Suggestion: A direct ranking comparison between statistical feature selection and deep learning feature attribution should be provided to validate the feature selection process.

**Detailed Comments:**

1-Analysis of Model Performance Without Manual Feature Selection
🔹 The deep learning model was trained only on pre-selected features instead of learning feature importance automatically. This could bias the model towards statistical assumptions and prevent deep learning from discovering hidden relationships.
🔹 Suggestion: Train an alternative version of the model without manual feature selection and compare its performance with the pre-selected feature model. If deep learning discovers additional useful features, this would justify revisiting the feature selection process.

2-No Sensitivity Analysis on Feature Selection Choices
🔹 The paper does not test how model performance changes when different feature selection strategies are applied.
🔹 Suggestion: Conduct a sensitivity analysis by training models with different feature subsets (e.g., using LASSO, recursive feature elimination, or deep learning-based feature selection) to evaluate how robust the model is to feature selection variations.

**Justification Of The Preliminary Rating:**

The paper presents a well-structured deep learning approach for liver fibrosis staging using CT-based radiomic features. It includes statistical feature selection, multiple datasets (MSD, NIH, UW), and Integrated Gradients for model interpretability. The methodology is clinically relevant, and results show strong performance.

However, key concerns remain. The manual feature selection process may have removed non-linear patterns that deep learning could have detected. The lack of a direct comparison between statistical and deep learning-based feature selection raises concerns about bias. Additionally, the small dataset size (only 19 patients per fibrosis stage) increases the risk of overfitting, and no sensitivity analysis was conducted on feature selection choices. Computational efficiency benchmarks are also missing.

If the authors validate on an external dataset, compare feature selection methods, and analyze feature sensitivity, the paper would be stronger. Until then, concerns about feature bias and generalization remain, leading to a weak acceptance recommendation.

**Questions To Address In The Rebuttal:**

1-Have the authors trained a model without manual feature selection, allowing deep learning to automatically determine the most predictive features?
If not, would they consider testing this to compare performance?
2-Have the authors conducted a sensitivity analysis where they test alternative feature selection methods (e.g., LASSO, recursive feature elimination, deep learning-based selection)?
3-Would a hybrid approach combining statistical filtering with deep learning refinement improve robustness?

**Special Issue:**

No

---

> ### Author Response · Authors · 2025-03-07
> **Thank you for your suggestions to improve the paper quality.**
>
> **R.1.1 [Weaknesses]** Suggestion: A model should be tested without manual feature selection
>
> **Answer:** Thanks for your comment. We kindly remind the reviewer that we have not conducted manual feature selection. All features were automatically identified through statistical analysis. Please refer to Section 2.5 "Statistical Analysis" in the revised paper for more details. Briefly, we used uni- and multi-variate logistic regression models in this work.
>
> 1. A deep classifier could not be trained with the limited sample size used in this work, which motivated the choice of logistic regression models.
> 2. For univariate analysis, a multinomial logistic regression model was fit for each of the 33 features (serum and CT imaging) individually. A p-value obtained from each model indicated the “predictiveness” of the feature. A rule-of-thumb p-value correction was done (0.05/33 = 0.0015) and a feature was automatically chosen if the p-value fell below 0.0015.
> 3. Using these features, a baseline multivariate regression model was trained.
> 4. From this baseline model, three other models were created: (1) without both serum and PVAT features, (2) with serum but without PVAT, and (3) without serum but with PVAT.
> 5. We also compared this multivariate model against another model called the “top-5 features” model. First, all 33 features were used to train a separate multivariate model. Then, an automated feature importance analysis was performed, and the top-5 features with the greatest log-odds ratios (coefficients) were automatically identified. Finally, these top-5 features were used to train the “top-5 features” multivariate model.
>
> --
>
> **R.1.2 [Weaknesses]** A direct ranking comparison between statistical feature selection and deep learning feature attribution should be provided
>
> **Answer:** Thanks for your comment. We are confused by the reviewer’s comment since we do not attempt to use deep learning for direct feature selection in our work. Briefly, only 480 patients from the UW dataset with different stages of fibrosis were used. This patient sample is not sufficient to train a deep learning model for staging fibrosis. Therefore, we derived explainable CT imaging biomarkers (e.g., liver and perivascular adipose tissue volumes). These biomarkers were used to train a multivariate logistic regression model to stage advanced fibrosis and cirrhosis.
>
> --
>
> **R.1.3 [Detailed comments]** Train an alternative version of the model without manual feature selection and compare its performance with the pre-selected feature model.
>
> **Answer:** We kindly remind the reviewer that we have not conducted manual feature selection. Please see R1.1 for additional details.
>
> --
>
> **R.1.4 [Detailed comments]** Conduct a sensitivity analysis by training models with different feature subsets.
>
> **Answer:** Thanks for your comment. For comparison, we trained another multivariate model called the “top-5 features” model with features identified through automatic feature selection. Please see R1.1 (#5) for additional details.
>
> --
>
> **R.1.5 [Rebuttal Questions]** Have the authors trained a model without manual feature selection, allowing deep learning to automatically determine the most predictive features? If not, would they consider testing this to compare performance?
>
> **Answer:** Thanks for your suggestion. We kindly remind the reviewer that we have not conducted manual feature selection. All features were automatically identified through statistical analysis. We have also trained another model called the “top-5 features model” for performance comparison. Please see R1.1 for additional details.
>
> --
>
> **R.1.6 [Rebuttal Questions]** Have the authors conducted a sensitivity analysis where they test alternative feature selection methods (e.g., LASSO, recursive feature elimination, deep learning-based selection)?
>
> **Answer:** Thanks for your comment. In the revision, we have conducted a sensitivity analysis by training another model called the “top-5 features model” for performance comparison. Please refer to Section 2.5 "Statistical Analysis" and R.1.1 for more details. Section 3.3 “Multivariate Results” contains performance comparison results.
>
> --
>
> **R.1.7 [Justification]** Computational efficiency benchmarks are also missing.
>
> **Answer:** In the revision, we have provided details about the computational time taken for inference with the nnU-Net model (~ 3 minutes) and the logistic regression model (~2 minutes).

---

> > ### Comment · Reviewer_SWNe · 2025-03-14
> >
> > (Other reviewer here) Although I understood that you did not do manual feature selection, your reply to R.1.1 was still interesting to read because it more clearly listed the trained classifiers than in the original paper.
> >
> > There is *some* manual choice, however, in the n=5 feature selection for the top-5 model.  Probably a reasonable n, and not an issue I have, but indeed not justified or discussed I believe.
> >
> > Then, in items 2/3 I wondered how many features passed that bar, and I also wondered (maybe depending on that unknown ratio of discarded features) whether it would have made sense to use that reduced feature set for item 5, the feature importance analysis, instead of all 33 features.
> >
> > I agree with the reply and justification to R.1.2 that you could not have trained a DL model with 480 samples.

---

> > ### Comment · Reviewer_U6Qz · 2025-03-14
> >
> > Thank you for addressing my questions and considering my concerns in your rebuttal. I have no further inquiries and will maintain my initial rating

---

> > ### Author Response · Authors · 2025-03-14
> > **Thank you for the additional comments to improve the paper quality**
> >
> > Thank you for bringing this point to our attention and helping us to clarify the paper even more. As mentioned in section 2.5 “statistical analysis” paragraph 3, we selected the top 5 features because of their absolute log-odds coefficients. Coefficients greater than 0.05 were chosen (not based on a p-value cutoff). The remaining features were below this threshold. We have added this justification to the same paragraph above.
> >
> > We clarify the features selected through statistical feature selection. Out of 33 features, 16 features were selected that included serum, PVAT, and other CT imaging biomarkers. These features were the same for both advanced fibrosis and cirrhosis. They included: (1) LSVR, (2) LSN, (3) spleen volume, (4-10) volume proportions of Couinaud segments except segment 7, (11) FIB4, (12) APRI, (13-15) normalized SD CT attenuation for the main, left, and right portal veins, and (16) fat volume at the right portal vein. This has been added to Section 3.3. "Multivariate results".
> >
> > Yes, feature importance based on log-odds coefficients can also be conducted from the above 16 features. However, since we are testing a different feature selection method, we choose to use all 33 features for consistency.
> >
> > Thank you for recognizing the limited sample size in our work and the value proposed by our contributions.

---

### Author Rebuttal · Authors · 2025-03-07

**Rebuttal:**

We gratefully thank the reviewers for their constructive comments to improve the quality of our paper. Changes to the manuscript are highlighted in purple font. The revised paper has also been uploaded with this rebuttal comment.

Summary of changes:

1.	Clarified the reference standard creation for the annotation of the portal vein and its branches.

2.	Elaborated on the use of logistic regression models and the rationale for their choice.

3.	Comparison of the multivariate model against a new “top-5 features” model to further assess the feature sensitivity.

4.	Minor typos and clarifications have been added as requested by each reviewer.

**Supporting Material:**

/attachment/17b7730d06a5bffcf4a7aa4d5421a684b1fade7d.pdf

---

### Meta-Review · Area_Chair_TKmC · 2025-03-23

**Recommendation:** Accept (Poster)
**Confidence:** 4

**Metareview:**

The paper receives mixed reviews (two positive and one negative). Although reviewer uFLW raises some valid concerns, all three reviewers agree that the paper is overall well/clear written, describes a complete pipeline to a clinical problem, and the hypothesis and evaluation are reasonably justified. Considering this paper is in application track, I recommend acceptance.